# Corporate Governance and Risk Management: Lessons (Not) Learnt from the Financial Crisis

Alessandro Gennaro [1,*] and Michelle Nietlispach [2]

1 Department of Economics and Management, "Guglielmo Marconi" University, 00193 Rome, Italy
2 Department of Legal and Political Science, "Guglielmo Marconi" University, 00193 Rome, Italy; michelle.nietlispach@gmail.com
* Correspondence: a.gennaro@unimarconi.it; Tel.: +39-339-3274025

**Abstract:** The paper aims to understand if and which lessons have been learned since the financial crisis of 2007–2008, highlighting the main deficiencies which still affect the corporate governance and risk management systems more than a decade after. A survey was performed by collecting the answers to 15 questions about corporate governance and risk management practices, given by a representative sample of 200 finance professionals (100 from the USA, 50 from Italy, 50 from the UK). The survey allows saying that corporate governance codes and risk management approach, even though improved and implemented over the past decade, still present problems in terms of principles or application. The results provide insights into how corporate governance issues are addressed and how financial institutions and regulators learn and adapt from a crisis. The paper also gives new perspectives on corporate governance, indicating where regulators need to focus on to rethink the governance mechanisms.

**Keywords:** financial crisis; corporate governance; risk management; remuneration; transparency





## 1. Introduction

The financial crisis 2007–2008 was a truly global crisis with various reasons or several aspects of failure causing it. Deficiencies in corporate governance and risk management systems were two major reasons. Corporate governance in financial institutions played a big part in the financial turmoil back in 2007–2008. It was not just about computational model failures but also the lack of proper corporate governance systems, risk management procedures, and board responsibilities. One might think that much has been learned from the financial crisis and that the regulators and the companies are doing their best to implement the lessons learned. But is this the case? Are there lessons really learned and implemented? This paper presents some evidence and insights about that.

The global financial crisis of 2007–2008 was subject to many different research and studies from scholars, practitioners, institutions, governments. Especially the causes of the global financial crisis were of high interest (Clarke 2010; Laeven et al. 2010; Lang and Jagtiani 2010; Tarraf 2010; United Nations Conference on Trade and Development 2010; Yeoh 2010). Many researchers and scholars see the main reason for the final collapse of financial institutions and the financial crisis—these collapses led to the global financial crisis as a triggering event—in the deficiencies in corporate governance, as well as in risk management and internal control systems. Other reasons are rather supplementary to those deficiencies or probably also resulting from them (Kirkpatrick 2009; Yeoh 2010; Fetisov 2009). Before the financial crisis 2007–2008, there was a broad interest in corporate governance, and there were already some studies about it, e.g., about the board efficiency (Band 1992). But the interest in, and therefore the focus on, corporate governance became even more with and after the financial crisis 2007–2008. Much has been written about what happened, and it is essential to understand why the financial crisis 2007–2008 happened and what led to it (Shafer 2013). The results of the different studies deliver important insights

into what went wrong and what lessons the various parties involved should learn. Most of the financial institutions in 2007–2008 had deficiencies in corporate governance. They did not spend the attention they should have spent on implementing corporate governance mechanisms, including a proper integration of risk management and internal control in governance systems. All of this is covered in the Literature Review: the failures are shown, the reasons indagated, the lessons that may be learned in the aftermath of the global financial crisis highlighted. Corporate governance codes and other kinds of regulations are on the rise since the financial crisis 2007–2008. New mechanisms of governance have been created, established, and applied over the past decade. But problems of implementation and practice remain.

Did the involved parties really learn from the financial crisis 2007–2008 to avoid making the same mistake twice or more? This is the main question, which the paper would like to address and stress, demonstrating that what has been learned out of the crisis is still not fully and effectively implemented. In this paper, the most important lessons learned from the financial crisis are summarized and presented, the research performed on corporate governance through a representative survey is described and discussed. The results and the conclusions from the research are presented to understand if the lessons from the financial crisis are learned or not. Through the study, new perspectives on corporate governance, risk management, remuneration systems, etc., are opened, which will give new details, insights, and guidelines for corporate governance. It is important to include various aspects of corporate governance lapses during the crisis, which will give further important information and help in creating normative implications for future reforms (Tarraf 2010).

The rest of the paper is organized as follows. A literature review on improving corporate governance and risk management in financial institutions after the financial crisis is presented in Section 2. Sample, dataset, and methodology are presented in Section 3, while in Section 4, there is the analysis and discussion of the survey results performed. Final remarks on the obtained results, limitations of the study, and future research suggestions are shared in Section 5.

## 2. Literature Review

Causes and consequences of the global financial crisis are recounted with a review of the existing literature on corporate governance failure in financial institutions during the global financial crisis.

Many researchers have studied the financial crisis with a focus on corporate governance (Fetisov 2009; Clarke 2010; Lang and Jagtiani 2010; Tarraf 2010). Many different reasons led to the financial turmoil in 2008. Corporate governance processes and procedures attributed to the bankruptcy of financial institutions and finally led to the financial crisis (Kirkpatrick 2009; Yeoh 2010). Many studies highlighted that risk management, board practices, remuneration system, transparency and disclosure norms were found the main lacking areas of corporate governance (Kumar and Singh 2013).

Fetisov (2009) stated that the decline in corporate governance standards before the financial crisis was ultimately the reason for the turmoil in 2008, leading to the financial crisis.

Clarke (2010) said that the financial crisis in 2008 was a crisis of corporate governance and regulations. Corporate governance and regulations did not get the necessary attention before the financial crisis and led to many financial institutions' bankruptcy and collapse.

United Nations Conference on Trade and Development (2010), in its analysis report on corporate governance in the wake of the financial crisis, stated that poor corporate governance practices are one of the reasons that caused the global financial crisis. In most of the collapsed financial institutions, they observed fragile and inferior risk management systems. Kirkpatrick (2009) observed that the financial crisis could be to a major extent attributed to failures and weaknesses in corporate governance. When the financial institutions were put under stress, the routines and procedures of corporate governance did

not quite serve the purpose of safeguarding against excessive risk-taking in many of the financial institutions (Kirkpatrick 2009).

Yeoh (2010) directed the attention of his research on several mistakes in transparency, disclosure norms, and the role of non-executive directors in financial institutions. He observed lacking commitment, lack of time, and competence to manage deals with complex financial products, which puts the financial institutions at high risk. Yeoh (2010) named Lehman Brothers as an example, as they were offering complex products and showed clear deficiencies in communication and transparency.

Lang and Jagtiani (2010) found a general and basic deficiency of the risk control system in most financial institutions. "Financial firms lacked effective internal controls, accurate and timely financial and risk reporting to the right management level, and a corporate-wide view of risk or an enterprise-wide risk management program" (Lang and Jagtiani 2010, p. 21). Moreover, Lang and Jagtiani (2010) pointed out that the boards of financial institutions failed to ensure that appropriate risk management systems are in place to address risk exposures to toxic financial products. The Authors concluded that the remuneration, incentive, and compensation systems of the financial institutions promoted an excessive risk-taking attitude. This motivated managers to increase the profitability of the business instead of focusing on the risk position of the enterprise. Lang and Jagtiani (2010) stated that there were also complex financial products such as CDOs and MBS, which were the ideal choice for executives as they generated huge revenue upfront, and they did not have to disclose the risk positions.

Laeven et al. (2010) suggested that complex financial products such as CDOs and MBS were accelerating the opaqueness in financial reporting to shareholders. The banks' incentive systems focused only on short-term goals and scenarios, resulting in excessive risk-taking by executives.

Pirson and Turnbull (2010) stated that directors and boards of corporate governance systems failed to monitor the risk management systems. They failed to control the excessive risk-taking behavior of management. Additionally, the non-executives did not raise questions on the deficiencies in the remuneration and incentive systems of the executives.

Blundell-Wignall et al. (2008, p.11) found that in 2008 that one of the major reasons for the financial crisis was the corporate governance in financial institutions. Often, there were insufficient risk control procedures, and the boards failed in providing strategic guidance. There was a general lack of understanding of enterprise risk in financial institutions as well as product risks. When offering complex financial products, the active management of product risks is of immense importance.

Berrone (2008) and Van Den Van Den Berghe (2009) indicated that the incentive systems with their common short-term targets for the executives of financial institutions played an essential part in the financial crisis 2007–2008. It is seen as one of the main reasons why it happened. Berrone (2008) highlighted that the remuneration and exit packages for executive employees were triggering wrong incentives and the question about moral hazard comes up.

Kirkpatrick (2009) found certain inherent deficiencies in the system itself. He stated that in nearly all of the failed financial institutions, the insufficient processes in corporate governance were related to the deficiencies in risk management and the risk management processes. Hence, there were no adequate risk management processes. Often, the board of the failed financial institutions did not suitably consider the risks when defining the company strategy. This led to the circumstance that foreseeable risk factors were not properly disclosed, and clearly, a system for properly monitoring and managing risks was missing. Additionally, Kirkpatrick (2009) directed attention to the misalignment or non-alignment of the financial institution's remuneration system to the institution's strategy, risk appetite, and long-term sustainability and goal.

Buiter (2009) found that the remuneration structure of professionals in the banking sector allowed them to take extreme risks with only a short-term focus.

Sahlman (2009) pointed out that " . . . many organizations suffered from a lethal combination of powerful, sometimes misguided incentives; inadequate control and risk management systems; misleading accounting; and low-quality human capital in terms of integrity and/or competence, all wrapped in a culture that failed to provide a sensible guide for managerial behavior" (p. 4).

Shafer (2013) stated that throughout the Great Moderation, there were several trends observable that led to the extraordinary fragile system before and during the global financial crisis 2007–2008. The trends can be summarized in four trends in financial markets: (1) rising leverage, (2) increasing maturity transformation, (3) increase in information asymmetries (more opaque financial instruments and markets generated by financial innovation), (4) higher intensity of incentive-based compensation in financial institutions.

Chen et al. (2019) discuss the economic recovery of the financial crisis 2007–2008, a decade later. Decisions about policies corporate governance before the financial crisis 2007–2008 had a high impact on the variation in output after the crisis. Emphasizing the importance of macroprudential policies and effective supervision, the countries that had greater financial vulnerabilities in the pre-crisis years suffered larger output losses after the crisis. Countries with a flexible exchange rate regime and countries with a strong fiscal position in the time before the financial crisis 2007–2008 experienced smaller losses. Countries that took postcrisis policy actions as a mitigation act showed fewer postcrisis losses.

Adrian et al. (2018) emphasize that regulatory reforms after the financial crisis 2007–2008 have provided a financial system that is more robust. However, the evidence of any unfavorable consequences is not clear until today. The consequences must also be seen in relation to other favors and benefits of regulatory reforms. Regulatory reforms can also reduce the probability of broad market liquidity crises and come with more robust market-making. International standard setters have an eye on evaluating the advantages and disadvantages, including the benefits and costs of new regulatory regimes. Further, one can expect a more granular quantification in the coming years. The evidence and findings of Adrian et al. suggest that the financial system has become safer through regulatory reforms at limited unintended cost.

Bhar and Malliaris (2021) highlight that the USA faced a long recession from the end of 2007 to mid of 2009, and the US Government and the Federal Reserve took expansive fiscal and monetary actions and policies to minimize the severity and duration of the recession. Remarkably was that the Federal Reserve established the "Quantitative Easing" policies, which took three rounds. The policies aim to reduce long-term interest rates in case the federal funds reach short-term rates of zero. The authors conclude that "Quantitative Easing" and regulatory reforms have contributed to the increases in the stock market's significant recovery since its crash due to the financial crisis.

The studies and literature on corporate governance failure during the financial crisis show various insights on how lessons learned from the crisis can design new policies or frameworks. The deficiencies that led to the financial crisis in 2008 were mainly in the corporate governance areas, including risk management, board practices, remuneration system, and shareholder rights.

Many researchers found the remuneration system as the key reason or key lesson from the financial crisis, which needs to be addressed with suitable measures for correcting those deficiencies of short-term scenarios (Bruner 2010; Laeven et al. 2010; Fetisov 2009; Lang and Jagtiani 2010; Van Den Berghe 2009). Huge risks can arise from the remuneration system when the focus is on short-term revenue and rewards managers for their failures. Academics have suggested linking the remuneration structure with long-term company performance (Laeven et al. 2010). Lang and Jagtiani (2010) pointed out that the remuneration system should also be linked to the risk appetite. Additionally, this should be within the internal controls and the risk management system (Lang and Jagtiani 2010). One of

the main trends that led to the financial crisis 2007–2008, according to Shafer (2013), is the higher intensity of incentive-based compensation in financial institutions.

As a second area of improvement, the risk management system was pointed out by many researchers (Tarraf 2010; Aebi et al. 2011). The key lesson for financial institutions from the financial crisis of 2008 is that broad risk management should be implemented by a Chief Risk Officer (Muelbert 2009). The fit for this position should have the rights and authority to implement a full risk management system under the board's control. Lang and Jagtiani (2010) demanded nearly the same reforms but focused more on implementing an effective internal control and risk management system. Additionally, Pirson and Turnbull (2010) suggested implementing effective internal control and risk management system but stressed the improvement of the risk control and oversight. Pirson and Turnbull (2010) mentioned and proposed a corporate network system that would help the board get regular feedback about different risk factors from different stakeholders. This enables them to also get the information at an early stage when a proactive action is still possible.

As a third area of improvement, many researchers pointed out the board composition and practices as well as the shareholder rights. Adams (2009) pointed out, that directors of financial institutions must have sufficient knowledge and expertise in finance which can also be ensured through development and learning and by hiring professionals bringing the skills needed (Adams 2009). Muelbert (2009) stated that it also makes sense to hire individuals with prior experience and expertise in risk management to deal with risk control and the firm's management. Adams (2009) shared that risk-taking can also be reduced by paying an adequate salary to directors to meet the difficulties and bear the responsibilities coming with their position.

As a last area of improvement, the reforming of the accounting standards and transparency and disclosure norms should be considered for reforms. Clarke (2010) proposed implementing a universal and global accounting and valuation method to ensure fair reporting to shareholders and investors. Kirkpatrick (2009) stated that, in general, there is a need for a new risk disclosure policy and a universal framework and guideline on risk management accounting.

It is visible that corporate governance gained much attention from many scholars, researchers, and analysts. Their studies often found that deficiencies in corporate governance practices played a huge role and contribution to the financial crisis 2007–2008. The topic is key, and corporate governance reforms and improvements can help prevent systematic, moral, and systemic failures within financial institutions.

The conclusion of the literature review is summarized as follows: board practices, risk management, remuneration system, and transparency and disclosure norms were insufficient in a variety of aspects.

## 3. Data and Methodology

In order to verify if the corporate governance codes and risk management approaches have been improved and implemented over the past decade, we performed a survey. The views and opinions of 200 professionals were collected through an anonymous questionnaire administered via SurveyMonkey in February 2021. To better evaluate the laws and regulations before and after the financial crisis and to highlight the lessons learned and if they were implemented or not, the survey questions were separated into six different areas: (1) principles of corporate governance (2) remuneration process (3) relevance of risk management (4) governance and risk management (5) board practices (6) shareholder rights. The questions are detailed in Appendix A.

The answers to the survey questions were analyzed to confirm or refute, for each area, the following research hypotheses.

- H1-Principles of corporate governance: most financial institutions have extended their corporate governance standards after the financial crisis 2007–2008, and there is no general need to revise the existing standards and/or to implement the existing ones better.

- H2-Remuneration process: most financial institutions have implemented long-term incentive and remuneration systems, defined explicit governance processes to establish remuneration, and to submit the remuneration policies to shareholder approval.
- H3-Relevance of risk management: most financial institutions implemented specific procedures to reach a broad integration of risk management into the business strategies definition so that the boards are well aware of the business risks.
- H4-Governance and risk management: most financial institutions adopt a company-wide approach to risk by implementing a clear risk management process and adequately disclosing risk data; therefore, there is no need to improve corporate governance codes in this area.
- H5-Board practices: being difficult to regulate board performances with binding law and regulations, most financial institutions implemented voluntary governance standards bettering the separation of functions in unitary boards.
- H6-Shareholder Rights: most financial institutions are already doing their best to support a constructive engagement with their shareholders, offsetting (at least in part) any need for further laws and regulations regarding the exercise of shareholder rights.

We defined the size of the respondent sample using Cochran's formula, based on the expectation that Governance Codes have been properly implemented, and financial institutions have widely achieved their application after the financial crisis of 2007–2008. Given the relevance of the crisis and the magnitude of its economic and financial effects, it was reasonable to expect a very strong response from regulators and financial intermediaries to ensure the stability and transparency of the financial system. Consequently, to consider the financial system response to the crisis adequate, we believe that improving the principles and standards in some specific areas of the corporate governance codes and their broad and effective application by financial firms were necessary. This explains our "strong" research hypothesis: at least 3 out of 4 respondents (75%) express a positive opinion about the codes' improvement, their implementation, and the risk management practices applied by the company they work for. Setting the threshold at 75% means, if reached, the US, UK, and Italian financial systems, almost in their entirety and with few exceptions, have benefited from a review of governance principles and standards and an improvement in the application of codes. A standard of 50% would mean that the simple majority, but not nearly all financial intermediaries, have improved their governance and risk management procedures, either voluntarily or because of the revision of the corporate governance codes. Evidence of this kind, even though it should be considered positively, does not seem to us a sufficient and consistent response to the financial crisis of 2007–2008.

To set a representative sample of finance professionals, we applied Cochran's formula, which allows researchers to estimate an ideal sample size given a desired level of precision, a confidence level, and an estimated proportion of the attribute present in the population. Cochran's formula is considered appropriate in situations with large populations. The confidence level in determining the sample sizes is usually 5% or 1%; it is already incorporated in the formula using the correspondent Z-value. The Cochran formula has been applied in several studies and various research areas; it has also been applied in finance (among others, Spengel and Wiegard 2011). Our research population is large as it includes finance professionals across Italy, USA, and UK. In Cochran's formula is a correction built-in that can correct and reduce the sample size. However, this correction is only possible if the population is relatively tiny (Cochran 1940; Cochran 2007). For having a representative study, the maximum margin of error the researcher is willing to accept falls between 4% and 8% at the 95% confidence level. Hence, 6% was applied to have representative research, which is within the named limits.

The following formula gives the optimal number of observations (n):

$$n = \frac{Z^2 \cdot p \cdot q}{e^2}. \tag{1}$$

where:

-      *Z*, the Z-value found in a Z Table, is set considering a confidence level of 95%,
-      *p*, the estimated portion of the population that has the attribute in question (positive thinking about the improvement of codes and managerial practices), is set at a level of 75%,
-      *q*, is $1 - p$,
-      *e*, the desired level of precision (i.e., the margin of error) is set at a level of 6%.

Consistent with Equation (1), the optimal sample size was 196 observations; so, the sample used for this research was 200 people. The sample consisted of 100 people from the USA, 50 from Italy, and 50 from the UK. In all countries, most interviewees were women (57% versus 43% considering the entire sample); British and Italian respondents were concentrated in the age group 30–44, while the American ones were less concentrated with a relative majority in the age group 45–60 (see Figure 1a,b).

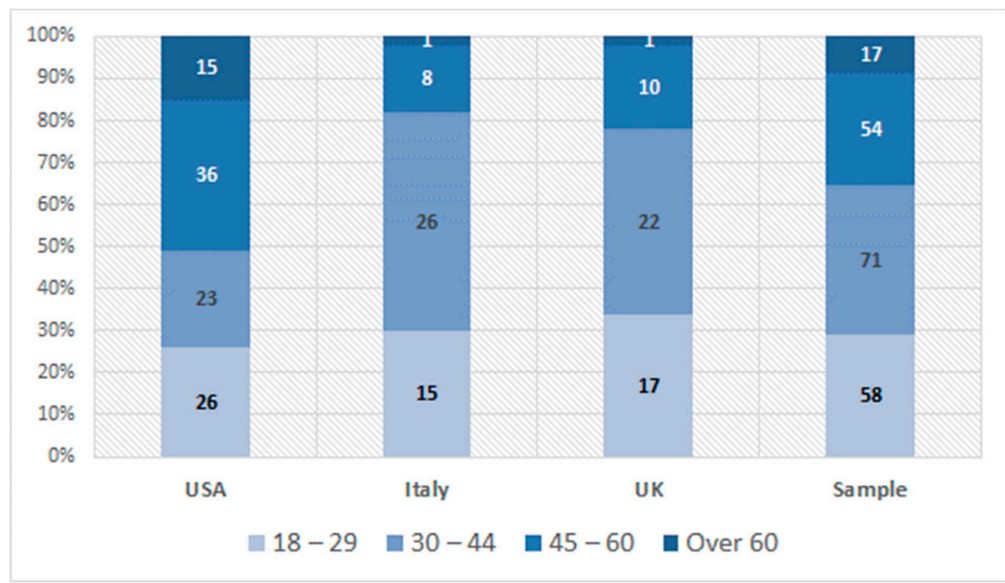

(**a**) Age of respondents by country

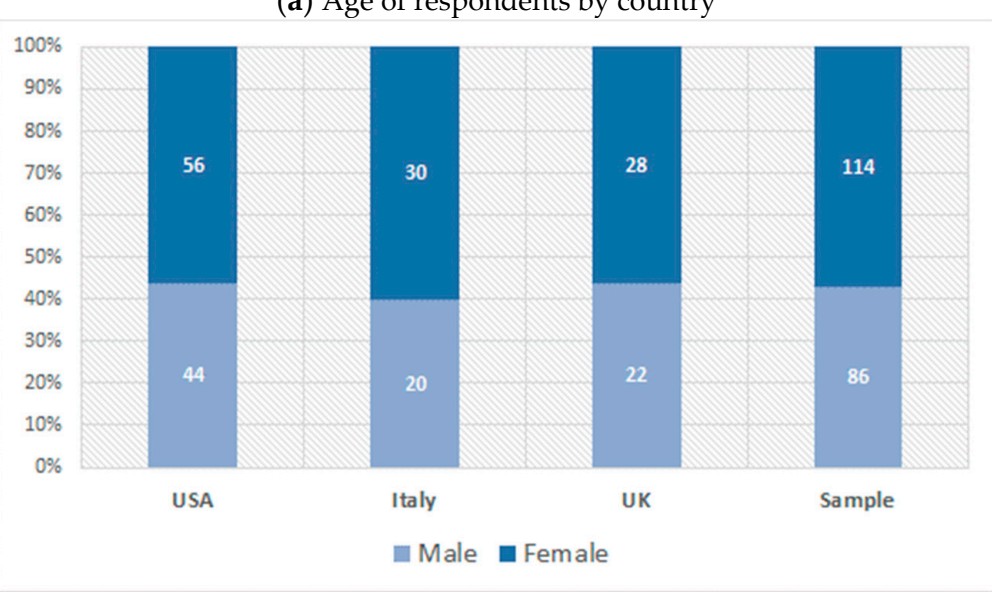

(**b**) Sex of respondents by country

**Figure 1.** Composition of the sample of respondents.

## 4. Results

The research questions were separated into six different corporate governance areas: (1) principles of corporate governance (2) remuneration process (3) relevance of risk management (4) governance and risk management (5) board practices (6) shareholder rights

*(1)    Principles of corporate governance*

Corporate governance in financial institutions played a big part in the financial turmoil back in 2008. It was not about computational model failures but the lack of proper corporate governance, including risk management procedures and board responsibilities. Our expectation was that more than a decade after the financial crisis in 2008, major lessons were learned, corporate governance, therefore, has a solid foundation in companies, and a sufficient legal basis was created with sufficient supervision. Therefore, our research hypothesis was that most financial institutions have extended their corporate governance standards after the 2007–2008 financial crisis. Since the questions provide only dichotomous answers (YES/NO) and were formulated in such a way that the answer "no" was related just to opinions of improvement of codes, our expectation was of more than 75% of the answer "no."

The survey showed that 70.5% of the sample responded "yes" to Q1, meaning there was a general need for revision of the existing standards or principles relating to corporate governance. Additionally, remarkable was the urgent need for implementation of yet existing standards or principles, which would need further enforcement to be properly effective (see Figure 2).

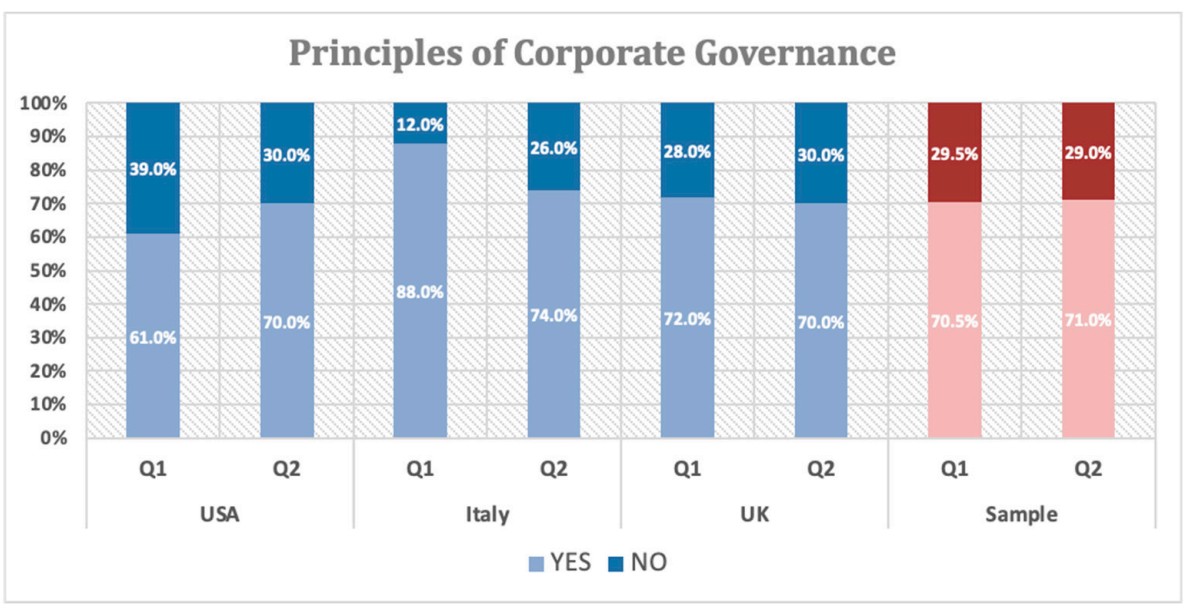

**Figure 2.** Answers to questions about the principles of corporate governance (1 and 2) by country of respondents.

Our research hypothesis was not confirmed by evidence and must be declined if the sample was considered as a whole. However, the conclusion was different when just focusing on Italy. The positive responses provided by the professionals who operate in Italy greatly exceeded the 75% threshold for Q1 and were very close to that threshold for Q2.

*(2)    Remuneration process*

Scholars widely studied changes in the remuneration process after the financial crisis 2007–2008. The expectation was to find a significant change from short-term oriented incentive and remuneration systems, which were widely spread in the financial crisis, to

long-term oriented ones. Hence, the questionnaire tried to indicate the improvements of the governance processes relating to remuneration and incentives (e.g., remuneration policies to be submitted to the annual meeting and, as appropriate, subject to shareholder approval). Therefore, our research hypothesis was that most financial institutions had implemented long-term incentives and remuneration systems. Since the questions were formulated so that the answer "yes" was related to opinions of improvement of remuneration process, we expected that more than 75% of answers were "yes".

The survey showed that about two participant companies in three (67%) were following an explicit governance process to establish remuneration, where roles and responsibilities of those involved, including consultants and risk managers, were clearly defined and separated (see Figure 3). In contrast, just one company in two (55%) comprehensively implemented the design of the long-term remuneration and incentive system. When it came to the question of whether the remuneration policies of the company were submitted to the annual meeting and, as appropriate, subject to shareholder approval; a bit more than half of the companies in the sample (59.5%) implemented such process steps. Put in relation to the three questions, it emerges that when a company follows explicit governance processes, normally, the design of the remuneration/incentive system is relatively long-term oriented. The remuneration/incentive policies are submitted to the annual meeting and are subject to shareholder approval. Hence, a significant tendency was visible: once the companies implemented the governance policies and processes, they applied them in a wide spectrum.

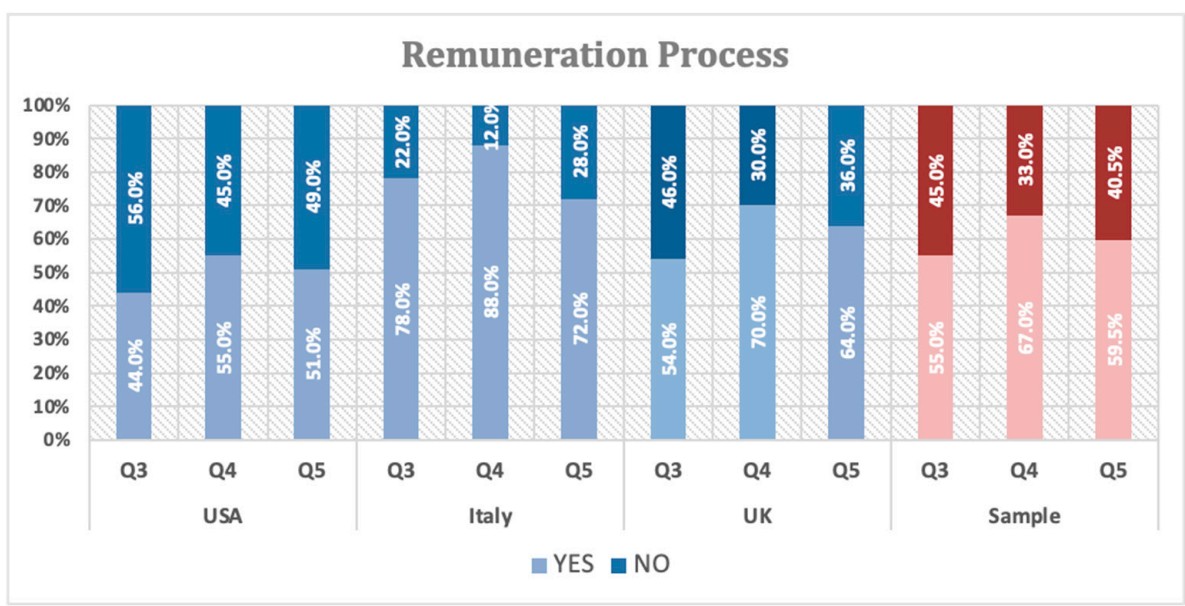

**Figure 3.** Answers to questions about the remuneration process (3, 4 and 5) by country of respondents.

Moreover, in this case, the research hypothesis was not confirmed by the entire sample, but the conclusion could change, focusing on Italy. Once again, the positive responses provided by the professionals who operate in Italy exceeded the 75% threshold for Q3 and Q4 and were very close to that threshold for Q5.

*(3)* Relevance of risk management

In this part, we expected a clear tendency toward a broad integration of risk management into the company's strategy so that boards are well aware of the business risks. Therefore, our research hypothesis was that most financial institutions implemented governance processes that integrate risk management into the company's strategy, making the board face and know about the risks.

Since the questions were formulated so that the answer "yes" was related to opinions of an increase in the relevance of risk management, our expectation was more than 75% of the answer "yes" to the question n. 7.

A bit more than half of the interviewees (55%) shared the opinion that the management and the board do not realize the importance of the integration of risk management into the company's strategy (see Figure 4). It meant that the board can be almost ignorant of the company's risks because risk managers are often kept separate from top management and are still not regarded as an essential part of implementing the company's strategy. It seems that the desired change did not happen comprehensively throughout all the financial institutions, a symptom of a lesson not learned. However, one respondent in two (51%) noticed a change in risk management and a bit more integration of risk management into the management/strategy after the financial crisis.

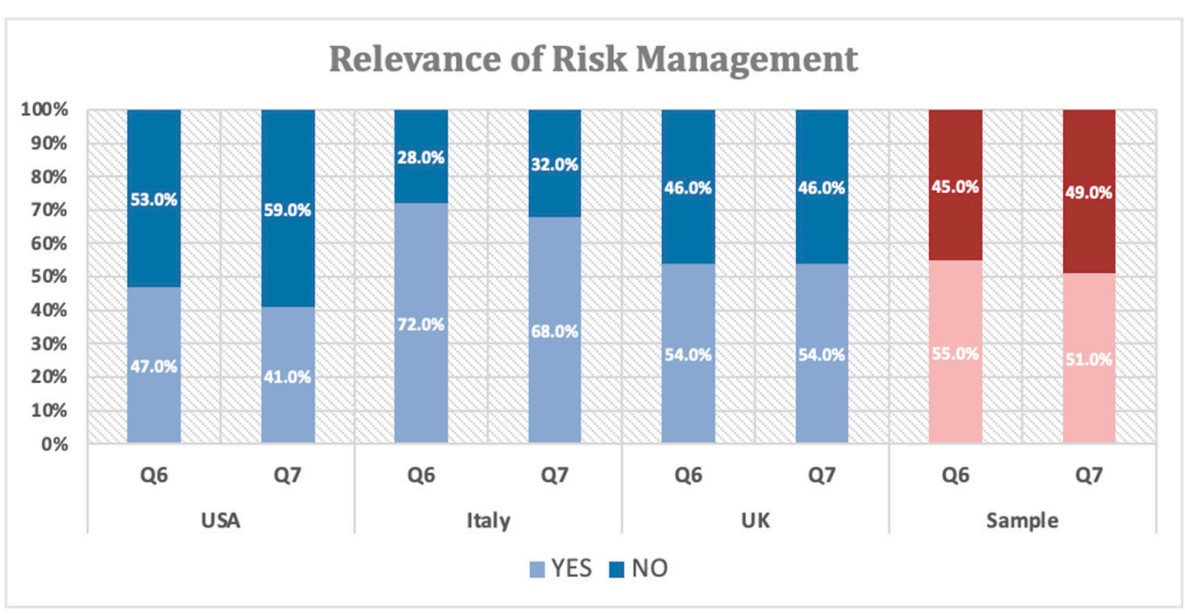

**Figure 4.** Answers to questions about the relevance of risk management (6 and 7) by country of respondents.

In this case, our research hypothesis was not confirmed by evidence and must be declined after observing the sample as a whole or considering a single country.

*(4)* Governance and risk management

Many studies in financial literature explained why the separation of the President and the CEO's roles improves corporate governance. Duality should send a positive signal to the market and improve the tendency to shareholder value maximization. If there is no obvious conflict of interest, a board could get to better and effective decisions. A clear tendency of an enterprise-wide risk management approach to implement a clear risk management process and properly disclosed risk figures was expected for this part of the research. Therefore, our research hypothesis was that most financial institutions implemented a clear risk management process according to an enterprise-wide risk management approach along with risk figures disclosed according to standards set in a code. The answer "no" to questions 8 and 9 relates to opinions of no need of integrating codes with references to risk management and risk disclosure; then, our expectation was of more than 75% of the answer "no" to these questions. Questions 10 and 11 were formulated in such a way that the answer "yes" was related to opinions of already well implemented "segregation duty" in a company; then, our expectation was of more than 75% of the answer "yes" to these questions.

The survey clearly showed another lesson not learned and an important problem not faced and solved: effective implementation of risk management requires an enterprise-

wide approach rather than treating each business unit individually. More than 70% of respondents said that (see Figure 5a):

- Corporate governance standard setters should be encouraged to include and/or improve references to risk management to raise awareness and improve implementation;
- There is a need for the process of risk management and the results of risk assessments to be appropriately disclosed and the related standards set in a code.

Moreover, 63.5% of the sample answered that the board is involved in both establishing and overseeing that the risk management structure in the company, and good practice is implemented. Still, risk management and control functions are independent of profit centers, and the Chief Risk Officer reports to the Board of Directors along the lines (see Figure 5b).

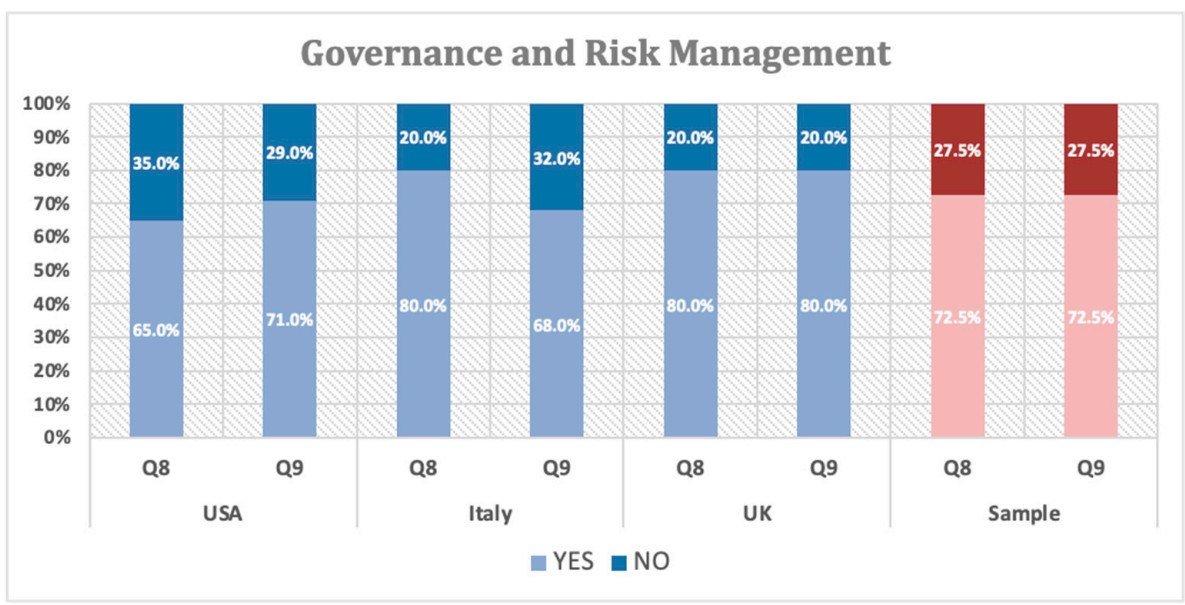

(**a**) Answers to questions 8 and 9

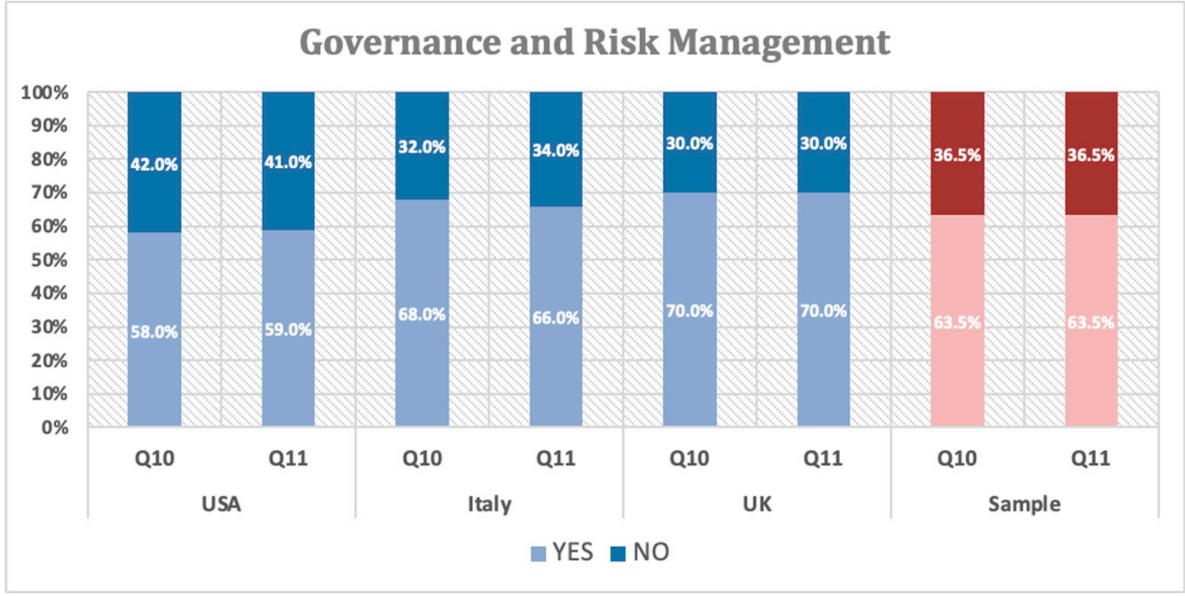

(**b**) Answers to questions 10 and 11

**Figure 5.** Answers to questions about the governance of risk management by country of respondents.

In this case, the research hypothesis was not confirmed and must be declined considering the sample as a whole, but the evidence of individual countries described rather different situations.

*(5)    Board practices*

It appears difficult and perhaps impossible to find a "silver bullet" in the form of laws and regulations to improve board performance. This leaves the private sector with an important responsibility to improve board practices through, inter alia, implementing voluntary standards. In this area, our research hypothesis was that most financial institutions implemented the separation of the functions of the Chief Executive Officer and the Chair of the Board of Directors in unitary boards. The answer "yes" to question 13 indicates a right implemented separation between CEO and Chairman; hence, our expectation was of more than 75% of the answer "yes" to this question.

A bit more than half of the sample (56.5%) thought this was not an acceptable status to leave the board practices as a voluntary standard without a binding law. When it comes to separating the functions between the Chief Executive Officer and the Chair of the Board of Directors, they are separated in most companies (63.5%); however, there were still 36.5% stating that the functions are not separated (see Figure 6). Moreover, the "comply or explain" and associated transparency are necessary to preserve companies' flexibility in special situations.

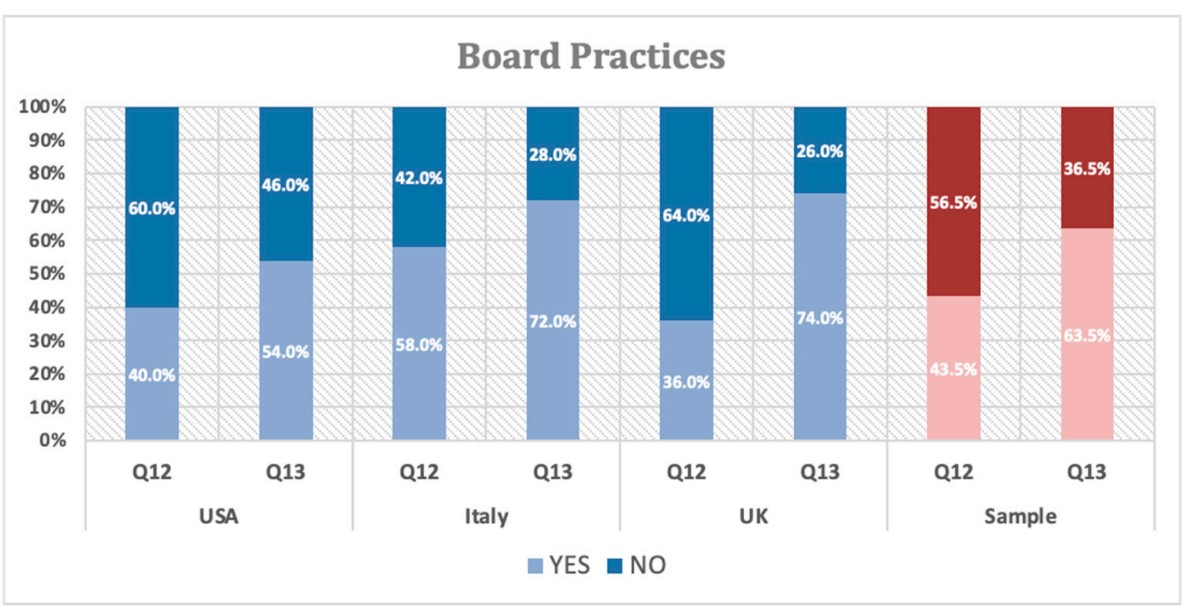

**Figure 6.** Answers to questions about the board practices (12 and 13) by country of respondents.

The research hypothesis must be declined considering the entire sample, but evidence from Italy and the UK showed that the threshold of 75% was close to being reached.

*(6)    Shareholder rights*

The interests of some shareholders and those of management are "aligned" in a period of a bull market. But this is not sustainable in the long run. In the past, it has been associated with a great deal of short-term behavior that has (badly) affected the operating and strategies of many companies. While there are different types of shareholders, they tend to be reactive rather than proactive and seldom challenge boards in sufficient numbers to make a difference. Therefore, our expectation was this: most financial institutions still face alignments of the interests of shareholder and management. Therefore, they expect company management to do more to support constructive engagement with their shareholders.

Questions 14 and 15 were formulated so that the answer "no" was related to opinions of adequate protection of shareholders' rights; therefore, we expected that more than 75% of the answers were "no".

In 78% of cases, the interviewed found room for improvement—companies need to do more, and it is in their interest to support constructive engagement with their shareholders. About three respondents in four (75.5%) demanded further laws and regulations for the exercise of shareholder rights (see Figure 7).

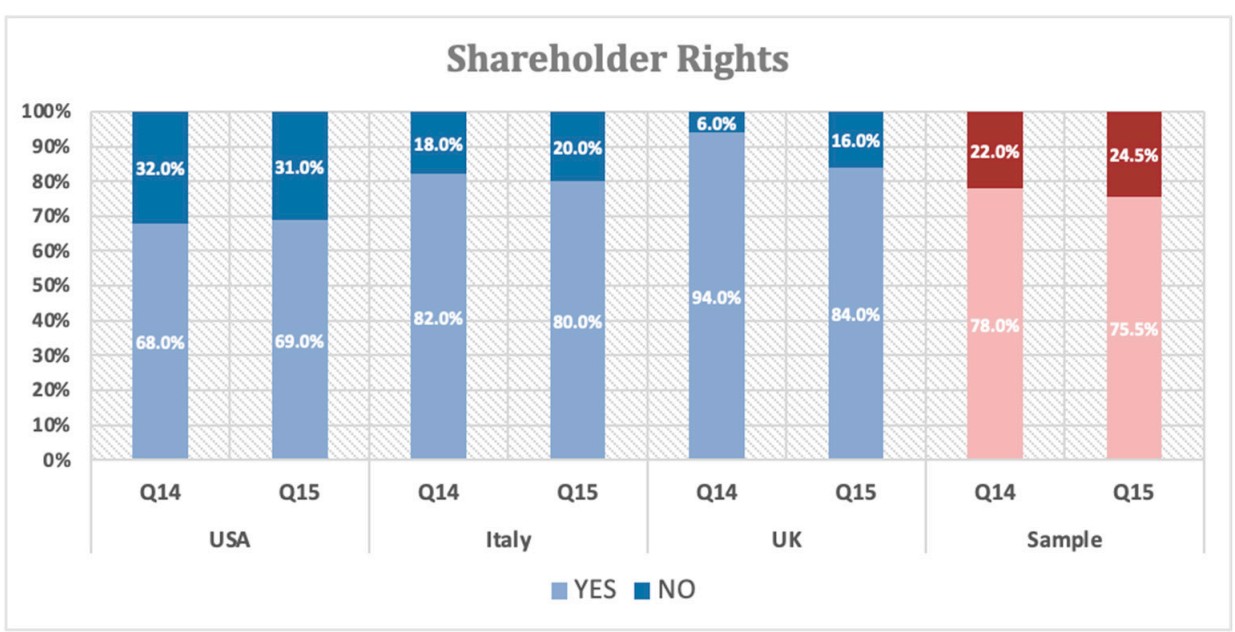

**Figure 7.** Answers to questions about the shareholder rights (14 and 15) by country of respondents.

In this case, our research hypothesis should be denied, as there was no evidence in any country included in the sample of an engagement or regulation sufficiently implemented.

Considering this evidence, it was important to check if the survey questions were interconnected to formulate consistent conclusions. We completed the research with a statistical analysis of the correlation between the answers to the 15 questions to highlight the extent to which respondents expressed related opinions on the improvement of the codes, their implementation, and the risk management practices applied by the company they work for. A correlation matrix was built (see Table 1) by considering:

-   The answers to the questions (yes = 1; no = 0) and sex (male = 0; female = 1) as dichotomous quantitative variables;
-   Age (18–29 = 1; 30–44 = 2; 45–60 = 3; over 60 = 4) and country (US = 1; Italy = 2; UK = 3) as discrete quantitative variables.

Since the answers to the questions, were dichotomous, a null correlation between two questions indicated that 50% of the respondents provided the same answer, and the remaining 50% had a different answer. A positive (negative) correlation between two questions and equal to 0.5 indicated that around 75% of respondents gave the same (a different) answer and only 25% a different (the same) answer.

**Table 1.** Correlation matrix.

| | Q1 | Q2 | Q3 | Q4 | Q5 | Q6 | Q7 | Q8 | Q9 | Q10 | Q11 | Q12 | Q13 | Q14 | Q15 | A | S | C |
|---|---|---|---|---|---|---|---|---|---|---|---|---|---|---|---|---|---|---|
| Question 1 | 100% | | | | | | | | | | | | | | | | | |
| Question 2 | 50.5% | 100% | | | | | | | | | | | | | | | | |
| Question 3 | 34.0% | 24.1% | 100% | | | | | | | | | | | | | | | |
| Question 4 | 22.2% | 30.1% | 34.8% | 100% | | | | | | | | | | | | | | |
| Question 5 | 31.5% | 28.1% | 40.0% | 39.6% | 100% | | | | | | | | | | | | | |
| Question 6 | 47.3% | 28.6% | 37.4% | 15.6% | 25.7% | 100% | | | | | | | | | | | | |
| Question 7 | 33.1% | 23.3% | 38.0% | 29.1% | 35.3% | 42.0% | 100% | | | | | | | | | | | |
| Question 8 | 36.3% | 42.1% | 27.6% | 21.1% | 35.9% | 36.6% | 18.0% | 100% | | | | | | | | | | |
| Question 9 | 31.4% | 44.5% | 11.8% | 23.5% | 26.7% | 23.1% | 20.3% | 54.9% | 100% | | | | | | | | | |
| Question 10 | 12.4% | 20.2% | 25.4% | 28.5% | 43.2% | 19.1% | 25.4% | 37.0% | 20.8% | 100% | | | | | | | | |
| Question 11 | 19.3% | 22.5% | 31.6% | 30.7% | 30.5% | 21.2% | 25.4% | 27.7% | 13.8% | 33.1% | 100% | | | | | | | |
| Question 12 | 17.0% | 22.7% | 24.6% | 35.8% | 27.2% | 32.7% | 33.6% | 17.9% | 8.9% | 26.7% | 26.7% | 100% | | | | | | |
| Question 13 | 35.2% | 22.5% | 35.8% | 24.1% | 34.8% | 33.7% | 31.6% | 37.0% | 27.7% | 28.8% | 37.4% | 18.3% | 100% | | | | | |
| Question 14 | 29.2% | 37.9% | 27.2% | 21.8% | 22.6% | 17.5% | 18.0% | 48.4% | 37.6% | 37.5% | 27.4% | 22.3% | 24.9% | 100% | | | | |
| Question 15 | 37.1% | 37.9% | 25.6% | 31.7% | 24.0% | 23.3% | 20.9% | 50.8% | 37.8% | 14.8% | 24.4% | 14.8% | 34.1% | 45.5% | 100% | | | |
| Age | 2.2% | −1.5% | −13.4% | −27.4% | −10.7% | −10.2% | −4.6% | 6.3% | 17.0% | −14.5% | −1.2% | −13.0% | 2.2% | 0.8% | −0.8% | 100% | | |
| Sex | 8.0% | 9.0% | 8.7% | 5.6% | 2.4% | 2.6% | −2.3% | 9.8% | 7.6% | 1.3% | −9.2% | −5.3% | 7.6% | 12.4% | 4.5% | −3.3% | 100% | |
| Country | 13.6% | 0.7% | 12.7% | 17.3% | 13.2% | 9.1% | 13.9% | 15.2% | 7.1% | 11.0% | 9.7% | −0.3% | 18.5% | 26.2% | 15.1% | −24.1% | 0.6% | 100% |

It can be noted that no negative correlations emerged and that correlation values were never relevant except between Q1 and Q2, Q1 and Q6, Q8 and Q9, Q8 and Q14, Q8 and Q15 when about 75% of respondents provided the same answers. The levels of correlation found were in line with our expectations and consistent with the literature review. It is useful to underline that the questions are almost always independent, but interesting interconnections emerge between:

- The need to revise existing standards or principles of corporate governance (Q1) and the need for a broader integration of risk management into the strategic planning and implementing (Q6);
- The lack of coverage of risk management principles or standards by existing corporate governance codes (Q8) and alignment between the interests of shareholders and the objectives of managers (Q14 and Q15).

The answers collected do not seem generally conditioned by respondents' age, sex, and country in which they operate. However, it can be observed that age influenced the answers to Q4 being the correlation index equal to 0.27 (the respondents with less than 45 years tend to answer yes, the older ones tend to answer no).

## 5. Discussion and Conclusions

The study aimed to demonstrate the lessons learned or not learned from the financial crisis of 2007–2008. Quantitative research was performed using a questionnaire containing 15 questions related to corporate governance and risk management practices. With a sample of 200 finance professionals (100 from the USA, 50 from Italy, and 50 from the UK), the research provides a basis to challenge and rethink how corporate governance issues are addressed and how financial institutions and regulators learn and adapt from a systemic crisis.

The survey makes it possible to reject all six research hypotheses relating to the corresponding areas of corporate governance in financial companies. The rejection of any research hypothesis means that the US, UK, and Italian financial systems, with some differences, have not benefited from a sufficiently comprehensive review of governance principles, standards, and procedures. Moving the attention from the entire sample to the individual countries, it emerges that the distance from the benchmarks confirms that the research hypotheses are sometimes very different between the USA, UK, and Italy. The answers provided by the Italian financial companies would make it possible to accept H2.

The choice of setting the threshold for positive or negative responses to each question at 75% is based on our strong conviction: the high magnitude of the 2007–08 crisis would have required a profound and extensive rethinking of governance practices in financial systems. We are aware that a lower threshold would have led to different conclusions. Lowering the threshold to 50%, which would also have required an increase in the sample size, would have allowed us to accept H2, H3, H5. The same threshold would not have

allowed H4 to be refuted or confirmed. Lowering the benchmark at 50% means that the simple majority (not the entire financial system with few exceptions) has improved their governance procedures in the areas of "remuneration process," "relevance of risk management," and "board practices."

The survey results and the rejection of the research hypothesis allow us to propose the following conclusions and suggest some perspectives on future corporate governance, highlighting the areas on which regulators and companies need to focus.

### 5.1. Principles of Corporate Governance

The financial crisis 2007–2008 was closely related to deficiencies in corporate governance and the insufficient implementation of corporate governance codes and principles within financial institutions. Policymakers, regulators, and authorities expected that more than a decade after the financial crisis in 2007–2008, major lessons were learned, and that corporate governance would have a solid foundation in companies and that a sufficient legal basis was created with sufficient supervision. Research results indicate that it is not the case. The wished and expected reforms did not happen. Some countries try harder than others and do revisions of their corporate governance standards. However, corporate governance standards are often voluntary, and good corporate governance is primarily the responsibility of every company. Hence, again, the management will follow their short-term rather than long-term goals, and corporate governance and sustainable business practices mostly play a subordinate role. In this area, lessons from the financial crisis seem not learned; companies still did not sufficiently extend their corporate governance standards. The implementation of corporate governance codes and principles in financial institutions remains insufficient after the 2007–2008. The problem seems to be still systematic.

Making corporate governance standards mandatory, both hard law and soft law should provide a comprehensive corporate governance framework, thereby encouraging the introduction of high governance standards and best practices in the companies' corporate governance system. Additionally, further supervision of corporate governance practices is needed, which is a consequence of what we said above.

### 5.2. Remuneration Process

The remuneration process of financial institutions and the whole remuneration governance are one of the reasons that led to the financial crisis 2007–2008. Therefore, the lack of a long-term incentive system and orientation on short-term goals is poisonous for a company. Major changes and reforms in the incentive systems would have been expected due to the financial crisis 2007–2008. Some improvement of the governance process relating to the remuneration and incentive system is visible as part of a general improvement of governance and processes. However, the most important part about the long-term remuneration and incentive system was only implemented by one company in two of the sample. Hence, one of the main lessons of the systemic crisis of 2007–2008 about the remuneration process seem not learned. The research indicates a need to include governance codes and principles "long-term oriented" for remuneration and incentive systems to make management act more sustainably.

### 5.3. Relevance of Risk Management

The research explains that today, many boards still ignore the importance of integrating risk management and competitive strategy. Due to the major impact that the deficiencies in risk management had and its part in leading to the financial crisis 2007–2008, major reforms and regulations would have been expected in this area after the financial crisis 2007–2008. But directors do still not comprehensively consider the risks that the company they manage really faces. Hence, the desired change did not happen extensively throughout all the financial institutions of the sample. Another lesson not learned. In this case, the results recommend considering the relevance of the implementation of risk management processes

into corporate governance standards. There is an urgent need to include principles and processes of risk assessment, risk management, and risk disclosure in the governance codes. This ensures that the board will know and accept the results the company faces under their leadership. Policymakers and regulators and supervisors are still not doing enough and ignore the risks or not properly addressing them.

### 5.4. Governance and Risk Management

Due to the financial crisis 2007–2008 and the analysis of the causes that led to it, major changes and reforms would have been expected to happen in the area of governance and risk management. Unsound risk management practices and complex products were a big issue and one of the reasons for the financial crisis 2007–2008. The expectation was that the corporate governance codes at least would include further references to risk management. But even with the knowledge gained from the 2007–2008 financial crisis, there are still not enough governance and risk management standards. It is visible from the research results that there is a need for further references to risk management in governance codes. The risk awareness is still not satisfying, and the risk management figures are still not disclosed in all the financial institutions. Here we can also observe a lack of binding codes to disclose the risk management results. Moreover, the effective implementation of risk management would require an enterprise-wide approach that is not yet applied in most financial institutions in the sample. A thin majority of the sample sees the board involved in both establishing and overseeing the risk management structure, the good practice implemented, the risk management and control functions independent from profit centers, the Chief Risk Officer reports to the Board of Directors along the lines. However, the desired change did not happen comprehensively throughout all the financial institutions to the full extent. The survey recommends including the proper references to risk management in the corporate governance standard and the regulation of the disclosure of risk management results and risk figures.

### 5.5. Board Practices

After the global crisis, we expected that most companies would implement the separation of the functions of CEO and Chairman as it was a major issue before the financial crisis 2007–2008. The segregation of the functions could be easily implemented without the tremendous investment of time and money. This lesson seems partially learned as nearly three-quarters of the sample companies have a proper separation of the functions. But what was not learned comes from the regulator side. The regulator did not include the Board Practices as "binding law" and mostly leaves the codes voluntary. The results highlight a need for binding corporate governance standards relating to board practices and the general separation of various functions. A "comply or explain" clause could be helpful here to let the companies remain flexible in special situations.

### 5.6. Shareholder Rights

The survey identified room for improvement regarding the exercise of shareholder rights: three respondents in four think the companies themselves need to do more and that there should be further laws and regulations that are binding. Therefore, lessons from the financial crisis seem to be not learned. There are still not appropriate and binding regulations implemented that would regulate the exercise of shareholder rights and avoid the alignment of interests. The results indicate that the exercise of shareholder rights should be defined and regulated in a mandatory corporate governance standard.

We are aware that two factors could influence the research results:

- Some responses fall in the critical range of 70%–74% and are not easily interpretable in a univocal way;
- Some questions may appear inclined to be answered in a way morally desirable rather than actually related to the company's governance problems.

In formulating and proposing our conclusions, we have tried to take these critical issues into account.

**Author Contributions:** Conceptualization, M.N.; methodology, A.G. and M.N.; validation, A.G.; formal analysis, A.G. and M.N.; investigation, M.N.; data curation, A.G.; writing—original draft preparation, M.N.; writing—review and editing, A.G.; supervision, A.G. All authors have read and agreed to the published version of the manuscript.

**Funding:** This research received no external funding.

**Data Availability Statement:** The results reported are supported by data collected through a survey conducted through SurveyMonkey.

**Conflicts of Interest:** The authors declare no conflict of interest.

## Appendix A

Sections and questions of the survey
*Section 1: Principles of corporate governance*

- Q1. Do you feel there is an immediate call for a revision of the existing standards or principles (e.g., OECD principles)?
- Q2. Do you think there is an urgent call or challenge to encourage and support effective implementation of already agreed standards regarding corporate governance in the country?

*Section 2: Remuneration process*

- Q3. Are there instruments implemented in your company that rewards an executive only once the performance is realized (i.e., ex-post accountability)?
- Q4. Does your company follow an explicit governance process to establish remuneration, where roles and responsibilities of those involved, including consultants, and risk managers, are clearly defined and separated?
- Q5. Are the Remuneration policies of your company submitted to the annual meeting and as appropriate, subject to shareholder approval?

*Section 3: Relevance of risk management*

- Q6. Perhaps one of the greatest shocks from the financial crisis has been the widespread failure of risk management. In many cases risk was not managed on an enterprise basis and not adjusted to corporate strategy. Risk managers were often kept separate from management and not regarded as an essential part of implementing the company's strategy. Most important of all, boards were in a number of cases ignorant of the risk facing the company: Is this still the case in your company?
- Q7. Did you experience a change in risk management and the integration of risk management into the Management/Strategy after the financial crisis?

*Section 4: Governance and risk management*

- Q8. With few exceptions, risk management is typically not covered, or is insufficiently covered, by existing corporate governance standards or codes. Do you think that corporate governance standard setters should be encouraged to include or improve references to risk management in order to raise awareness and improve implementation?
- Q9. With few exceptions, risk management is typically not covered, or is insufficiently covered, by existing corporate governance standards or codes. Would you encourage that the process of risk management and the results of risk assessment should be appropriately disclosed, and the standards set in a code?
- Q10. Effective Implementation of risk management requires an enterprise-wide approach rather than treating each business unit individually. Is the Board involved in both establishing and overseeing the risk management structure in your company?
- Q11. To assist the board in its work, it should also be considered good practice that risk management and control functions be independent of profit centers and the "chief risk

officer" or equivalent should report directly to the Board of Directors along the lines (already advocated in the OECD Principles for internal control functions reporting to the audit committee or equivalent). Is the "segregation of duties" as described above already implemented in your company?

*Section 5: Board Practices*

- Q12. It appears difficult and perhaps impossible to find a "silver bullet" in the form of laws and regulations to improve board performance. This leaves the private sector with an important responsibility to improve board practices through, inter alia, implementing voluntary standards. Do you think it is acceptable to leave the Board Practices as a voluntary standard without a binding law?
- Q13. When a dual board structure exists, the head of the management board should not become chair of the supervisory board upon retirement. In both cases, some form of "comply or explain" and associated transparency is necessary to preserve flexibility for companies in special situations. It should be considered good practice that the functions of Chief Executive Officer and Chair of the Board of Directors in unitary boards are separated. Is the function of Chief Executive Officer and Chair of the Board of Directors in your company separated?

*Section 6: Shareholder Rights*

- Q14. The interests of some shareholders and those of management have been "aligned" in the past period of a bull market but this was not sustainable and was associated with a great deal of short-term behavior. While there are different types of shareholders, they have tended to be reactive rather than proactive and seldom challenge boards in sufficient number to make a difference. Do you think that companies need to do more-and it is in their interest–to support constructive engagement with their shareholders?
- Q15. The interests of some shareholders and those of management have been "aligned" in the past period of a bull market but this was not sustainable and was associated with a great deal of short-term behavior. While there are different types of shareholders, they have tended to be reactive rather than proactive and seldom challenge boards in sufficient number to make a difference. Do you think that there should be further Law & Regulations regarding the Exercise of Shareholder Rights?

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
