# Peer review of "Corporate Governance and Risk Management: Lessons (Not) Learnt from the Financial Crisis"

_jrfm, doi:10.3390/jrfm14090419_

Round 1
Reviewer 1 Report
As the author stress the paper aims to understand if and which lessons have been learnt since the financial crisis of 2007-2008 highlighting the main deficiencies which still affect the corporate governance and risk management systems more than a decade after. In my opinon the research are quite interesting and address an important issue. Authors designed and conducted the study properly and drew appropriate conclusions. In my opinion, the article lacks a deeper indication of the causes of the situations and a fuller discussion of the presented results.
Author Response
Dear Referee,
thank you for your analysis, we really appreciate your comments and tried to improve our paper considering what you wrote in the review report.
Particularly, we improved some comments and conclusions.
Please see the attachement and the revised release of the paper for details.
Kind regards,
Alessandro.

Reviewer 2 Report
The article presents an interesting topic. The literature review is well done. The authors use in the research a questionnaire for which they demonstrate the assurance of the representativeness of the sample. However, the collected data are insufficiently capitalized. The authors limit themselves to the centralization of the answers, and the graphic representation of their distribution on the two options: "Yes" or "No".
I consider that the article can be improved, given the following:
- Clear definition of research hypotheses. Research hypotheses are not explicitly presented. These appear indirectly from the presentation of the results on the 6 areas of corporate governance. I consider that the inclusion of research hypotheses in the "Data and Methodology" section gives consistency to the work done.
- The graphics are repetitive and take up a lot of space. It would be useful to reorganize them. Perhaps, the presentation of a single graph for each area of corporate governance.
- Most of the times the answers to the questionnaire are in the median area (60% - 40%) between the two variants (Yes to No), which makes it difficult to formulate a clear conclusion, leading to the invalidation of all research hypotheses. At the same time, it is important to know if the answers to the 15 questions are interconnected. I consider it necessary to complete the research with a statistical analysis of the correlation between the answers to the 15 questions, in order to quantify more precisely the extent to which the respondents express a positive opinion about the improvement of the codes, of their implementation, and of the risk management practices applied by the company they work for.
Author Response
Dear Referee,
thank you for your analysis, we really appreciate your comments and tried to improve our paper considering what you wrote in the review report.
Particularly, we revised the presentation of the results and improved some comments and conclusions.
Please see the attachement and the revised release of the paper for details.
Kind regards,
Alessandro.

Reviewer 3 Report
Sorry for my late review. I don't think it was a paper to be reviewed for this long. There are no problems in this paper that need to be particularly corrected, and the survey results, even as it is now, would be worthy of publication.
The opinions of the reviewer below do not need to be taken into account in the revision of the manuscript.
- It seems possible to point out that there is no inevitable reason for the 75% standard. If you weren't expecting a sufficient level of improvements, you could lower the threshold to 50%, which would increase the sample size you need to collect. The survey results were at the level of 60-70%, and criticism may be raised that there is no compelling reason to conclude that there have been not enough improvements because they failed to meet the 75% threshold.
- Ages vary by country, with the 45-60 age group in the US and the 30-44 age group in Europe the most. This means that the positions within the companies of those who responded to the survey differ from country to country. They may have a different view of the company, and I think that this may be the reason for the differences between countries.
- Some questions may be particularly inclined to be answered that way because it is morally desirable rather than because the company has such problems. In other words, there may be some response levels that can come out even if you don't feel a problem with your company. Some people can agree that already good governance should be better.
Author Response
Dear Referee,
thank you for your analysis, we really appreciate your comments and tried to improve our paper considering what you wrote in the review report.
In particular, we have better explained the reasons for some methodological choices, and improved some comments and conclusions.
Please see the attachement and the revised release of the paper for details.
Kind regards,
Alessandro.

Round 2
Reviewer 2 Report
The article can be accepted in present form.